# Simulation Study of a Frame-Based Motion Correction Algorithm for Positron Emission Imaging

**DOI:** 10.3390/s21082608

**Published:** 2021-04-08

**Authors:** Héctor Espinós-Morató, David Cascales-Picó, Marina Vergara, Ángel Hernández-Martínez, José María Benlloch Baviera, María José Rodríguez-Álvarez

**Affiliations:** 1Instituto de Instrumentación para Imagen Molecular (i3M), Centro Mixto CSIC—Universitat Politècnica de València, 46022 Valencia, Spain; dcaspic@i3m.upv.es (D.C.-P.); maverdia@i3m.upv.es (M.V.); anher12a@i3m.upv.es (Á.H.-M.); benlloch@i3m.upv.es (J.M.B.B.); mjrodri@i3m.upv.es (M.J.R.-Á.); 2Department of Imaging and Pathology, Division of Nuclear Medicine, KU Leuven, 3000 Leuven, Belgium

**Keywords:** motion correction, multi-frame, spatio-temporal registration, simulation study

## Abstract

Positron emission tomography (PET) is a functional non-invasive imaging modality that uses radioactive substances (radiotracers) to measure changes in metabolic processes. Advances in scanner technology and data acquisition in the last decade have led to the development of more sophisticated PET devices with good spatial resolution (1–3 mm of full width at half maximum (FWHM)). However, there are involuntary motions produced by the patient inside the scanner that lead to image degradation and potentially to a misdiagnosis. The adverse effect of the motion in the reconstructed image increases as the spatial resolution of the current scanners continues improving. In order to correct this effect, motion correction techniques are becoming increasingly popular and further studied. This work presents a simulation study of an image motion correction using a frame-based algorithm. The method is able to cut the acquired data from the scanner in frames, taking into account the size of the object of study. This approach allows working with low statistical information without losing image quality. The frames are later registered using spatio-temporal registration developed in a multi-level way. To validate these results, several performance tests are applied to a set of simulated moving phantoms. The results obtained show that the method minimizes the intra-frame motion, improves the signal intensity over the background in comparison with other literature methods, produces excellent values of similarity with the ground-truth (static) image and is able to find a limit in the patient-injected dose when some prior knowledge of the lesion is present.

## 1. Introduction

Positron emission tomography (PET) is a medical imaging technique that measures tissue metabolism by following the body response of a tracer labelled with a positron emitter that travels along a specific metabolic path. Usually, PET scan acquisitions take between 20 and 90 min, and the patient must remain in exactly the same position during the whole scan time. However, in practice, patient motion is unavoidable. This is an important source of image degradation and artifacts [1,2] and a major source of delay in intrinsic spatial resolution improvements [3]. Although some methods can be applied to decrease patient motion, such as the use of head restraints [4,5], the motion correction algorithm approach is used to detect, track, estimate and correct this effect instead of preventing it.

Motion correction algorithms are usually based on an estimation and parametrization of the motion in order to include that information in the data prior or after the image reconstruction process. We call the former “online” methods and the latter “offline” methods.

An example of an extensively used online method is the optical motion tracking system in which we use a system that physically tracks and registers patient motion. Most recent optical motion tracking systems [6,7,8] enable online correction for motion that occurs within frames (quasi-static subsets of motion affected raw data) (intra-frame patient motion), but this method cannot be used in some cases, as when the view of the patient in the gantry is limited. In these cases, a frame-based motion correction method seems to be a better choice. Existing frame-based methods use the correlation coefficient [9], cross-correlation [10], sum of absolute differences [11], mutual information [12], mean square difference [13], standard deviation of the ratio of two images [11,13], stochastic sign change [13], or (scaled) least-square difference images [14,15], among others. Although frame-based motion correction methods do not have the same advantages as online optical motion tracking systems, they are very useful when old data need to be reanalyzed, when the data are not in a list-mode format, or when optical tracking approaches cannot be implemented due to a limited view of the PET gantry.

Once the motion is parametrized by the above-mentioned algorithms, that information needs to be included in the data in order to correct for motion artifacts. Here, there are two main procedures: image registration [16,17,18,19,20] and optical flow [21,22,23,24,25,26,27,28]. Whereas image registration methods aim to maximize the similarity between two images by finding and applying the optimal spatial transformation to one of them, the basic idea behind optical flow is to keep track of the spatial position of the image elements.

Both methods can be applied using some of the most used strategies listed below:Optimal gating [29,30] This defines the maximum time phase where the motion is minimal. A histogram is formed with the signals acquired during the PET scan and transformed into a cumulative distribution function. The lower and upper levels of this function settle the total sensitivity to a determined percentage (optimal gating yield).Motion compensated reconstruction [31,32,33] Here, motion information is integrated into a latter image reconstruction using the whole data set. In addition, this method implies using time varying system matrices.Joint reconstruction [34,35,36,37] This estimates the motion simultaneously with the image reconstruction. Therefore, only one image with full statistics is reconstructed. The high computational cost is a huge drawback of this approach.System matrix modelling [38] This technique uses the data in sinogram space and imposes a fixed transformation field from the information derived from the PET/CT instead of independently reconstructing the individual frames and the summing all of them up in image space, resulting in an improvement to the noise characteristics in the final images. This approach has some disadvantages: considerable uncertainty for non-rigid motions, its computational load, and the need to access the system matrix, which sometimes is not possible [39]. However, the potential for progress in this area in the coming years seems quite high, following the improvements and more common usage of GPUs [40] and third-party solutions integrated with their acquisition systems [41].Techniques based on projections and image deblurring [42] Among the techniques based on image deblurring, those using image deconvolution are the most used. It is a computational technique that allows one to compensate for the distortion of the image created by an optical system (in this case, PET). As is well known, all deconvolution techniques inherently amplify noise, so deconvolution is generally not an optimal strategy. Used conservatively, however, it has interesting results [43,44,45].Event rebinning [31] This method estimates the motion and transfers the information to the spatial coordinates of each line of response (LOR), rebinning them to a new position accordingly to the patient motion and then reconstructing the image statically. In this method, only affine transformations are allowed; therefore, it is not convenient for cardiac or respiratory motion. Another drawback of this method is that a non-negligible fraction of the events is lost as a result of some LORs that are rebinned outside the field of view (FOV) of the scanner. This fraction loss varies in practice depending on the nature and extent of the motion, and it leads to an increase in noise in the final reconstructed image.Image alignment (frame-based motion correction) [46,47,48] This consists of reconstructing each frame individually and aligning them to one reference frame. Although this method may produce images with a poor signal-to-noise ratio (SNR), this issue is overcome by averaging the aligned images. This technique is usually used in head, cardiac, or respiratory imaging procedures [49], as they are able to cope with non-periodic and non-rigid motions. This method is certainly the most used method nowadays together with event rebinning. An example of this technique is the well-known algorithm *multiple acquisition frames* (MAF) [50]. In the MAF method, the PET data are sorted into time frames, which are reconstructed separately, and then transformed using the information provided by a tracking system.

In a frame-based method, once all the frames are individually reconstructed, a final image is given by aligning and registering all the frames to a reference static or quasi-static image. In the literature, different methods can be found: Penney makes a comparison between various measures of similarity [51]. Sotiras focuses on the review of deformable medical image registration methods [52] that classifies the methods based on the central components of the registration—(i) deformation models, (ii) coincidence criteria, and (iii) optimization. Holden developed non-rigid geometric transformation models [53]. Other authors have focused their studies on specific anatomical regions, for example, cardiac imaging [18], or on techniques used for specific purposes, such as functional brain imaging [54], optical breast imaging [55], minimally invasive surgical procedures [56], or coronary heart disease [57]. From the point of view of the use of computational resources, we highlight the studies by Ramírez et al. [58].

Table 1 and Table 2 provide a brief summary of the different software packages most commonly used in medical image registration and their main characteristics.

Frame-based methods present some advantages when compared to other methods. For example, they can correctly estimate non-rigid motion in contrast to event rebinning. Joint reconstruction and system matrix modelling are excellent methods, but their computational load is quite high in comparison to that of the frame-based method. Lastly, in some cases, the system matrix is not available, so methods like motion compensated reconstruction or system matrix modelling simply cannot be applied. However, despite frame-based motion correction methods presenting a considerable amount of advantages, they also have deficiencies. One of them is the presence of noise in the reconstructed images when motion occurs inside the time duration of a given frame (intra-frame motion). Another example is the image degradation as a consequence of low count statistics that some of the frames can present when the motion is sudden. These deficiencies can affect the clinical practice as in the case of [59], where it is necessary to have frames with high statistics and low noise to improve lesion detectability. In order to overcome the greatest deficiencies of the MAF algorithm, a novel frame-based motion correction algorithm is proposed in this study. This method is called EMAF (enhanced multiple acquisition frames). Differently from MAF, this method allows the use of an adaptive threshold in order to divide the original dataset in frames when some prior knowledge of the size of the lesion is present, allowing one to reduce the dose administered to the patient while providing a good, motion-free estimation. It also allows for grouping motions that share similar features under the same frame, thereby increasing the statistics on each frame and reducing the intra-frame motion artifacts. Once the frames are defined, multilevel spatio-temporal registration is performed to ensure a good signal-to-noise ratio.

## 2. The EMAF Algorithm

The presented EMAF algorithm (shown in Figure 1) is an improved modification based on Picard’s original MAF Algorithm [50] in order to overcome their most important limitations.

In this method, the coincidence data given by the PET scanner are sorted into different frames using the information of an optical tracking system. Each of those frames are reconstructed separately, and, finally, an image registration method is performed to align and register the frames into the final reconstructed image. In order to clearly describe the algorithm, its explanation is divided into two different sections: first, the motion estimation in Section 2.1, and then the motion correction method in Section 2.2.

### 2.1. Motion Estimation

Once motion affected PET data are recorded and available, the EMAF algorithm firstly divides it into quasi-static frames. This is achieved by the following procedure:The motion is parametrized by the chosen tracking method.Using the motion given by the previous step, a region of interest around the source is defined. This region needs to be wide enough to fit any position recorded by the tracking method.The region of interest is rebinned into an equidistributed and equispaced grid in which the size of each bin (henceforth voxel) is a free parameter. The impact of this parameter is studied throughout this paper.Each of the voxels that make up the grid is numbered with an index associated with a certain position in the 3D space.Using the data of the tracking system, the voxel that contains the source in each moment is stored. This is achieved through two parameters, the time in which the source travels from one voxel to another (called cut-off time) and the index of the voxel (cut-off rates).The total time that the source spends in each voxel is calculated in order to apply a time filter. These steps are only followed for those voxels with a total time higher than half the average total voxel time. In this way, using voxels with small statistics that can add noise to the final image is avoided.For every voxel that passes the previous filter, a new list-mode file is created using the cut-off time and cut-off rate.When the process concludes, voxels that are completely empty are discarded.

At this point, it is worth mentioning that voxel size ratio selection is a relevant part of the algorithm. A large voxel size provides non-static frames, which ends in a deficient registration. On the other hand, a considerably small voxel size implies a large amount of frames with low statistics, which also has a negative effect on the final motion corrected image. It is important to note that the correct voxel size will always depend on the source size (Figure 2).

Statistical fluctuations in the position can lead to the classification of coincidences in an incorrect frame. This results in several frames outside the true trajectory of the source. Normally, these frames come with a small number of coincidence events. In order to solve this problem, a filter is used to delete any frame with a smaller number of coincidences than half the average.

Regarding the creation of the aforementioned grid, there can be two possible issues: (1) few frames may contain mixed motions and a blurring of the image in the registration phase, or (2) too many frames with poor statistics may cause the same consequences. Optionally, a minimal frame duration threshold (MFDT) can be set to limit and control the computation demands and time of the algorithm.

The approach exposed has two clear advantages: (1) it allows one to group in the same frame motions occurring in the same spatial region, which increases the statistical information, and this is one of the main drawbacks of the original MAF version [50]; (2) it does not need to prefix a priori the number of possible frames, which allows one to obtain a greater or lesser accuracy depending on the other voxel size of implementation being used.

### 2.2. Motion Correction

The motion correction registration EMAF step is a purpose-built subroutine written in C++ using the Insight Toolkit (ITK) architecture [60,61].

Normally, most studies use only the spatial information in the PET images, registering the images handled separately and treating them as though they are part of a discontinuous process [31,62]. In the EMAF algorithm, the temporal information is incorporated into a B-spline deformable registration in order to improve the registration accuracy based on the method developed for registering 2D ultrasound images [63], with adaptations to the registration of 3D PET images based on the developments made by Bai and Brady [64].

Given a reference and a misaligned image, the so-called reference and test image, respectively, the EMAF algorithm registers each of the test images to the reference image so that the motion between the different gates is estimated. The images are considered to be aligned by the time that the metric reaches the minimum value within a user-specified spatial precision. As long as the algorithm attempts to reach this value, it iterates and performs smaller transformations to the moving image each time until the required spatial resolution is eventually matched and a spatio-temporal B-spline is applied, as described in [64], which simultaneously registers all of the test images to the reference image. The precise variation of the transformation parameters at each step is controlled by the optimizer.

The cost function is minimized using gradient descent [65] with 0.01 mm spatial resolution, although other options are also implemented (for example, mutual information). The optimization is terminated when either a minimal incremental improvement in the cost function is achieved or after a pre-set maximum number of iterations defined by the user is reached. The registration step set-up is the rather standard root mean square error (RMSE), defined in Equation (Equation 1)
(1)$${\mathrm{RMSE}}_{A,B}=\sqrt{\frac{1}{n}\sum_{i=1}^{n}{\left(a_{i}\;-\;b_{i}\right)}^{2}}$$
where *a* and *b* stand for voxel intensity values of the respective images at a voxel index *i*, respectively, and *n* is the total number of voxels.

Following this process, it is shown that the spatial–temporal registration produces more accurate motion estimation than that of spatial registration only.

One of the points that is taken into account in the development of the registration part is the problem of tissue compression and the partial volume effect (PVE) since they lead to intensity modulations at the same points (pixels) of different reconstructed frames. This effect is usually more visible in thin structures. In order to avoid image defects caused by intensity variation, we consider the mass preservation property of PET images. We justify this, taking into account that, when the data are divided into frames, all of them are formed in the same time interval, that is, the entire acquisition time. In other words, in any frame, it is assumed that no radioactivity can be lost or added, apart from some minor changes to the edges of the field of view. For mass preservation, the variational algorithm for mass-preserving image registration (VAMPIRE) algorithm is used [66,67,68,69,70,71,72].

Finally, optimization of the multilevel registration algorithm is incorporated. First, the algorithm attempts to solve discrete optimization at a very coarse level (with a small number of unknowns.) Then, the approximate transformation is interpolated to the next finer level and used as an initial estimate for minimization of the next objective function. The procedure is repeated until the transformation is of the desired resolution. Multiple versions of the original images are constructed by iterative downsampling and smoothing of the data. This scheme has several advantages. First, the likelihood of ending in a local minimum is reduced, as only the main characteristics are taken into account at a certain level. Second, numeric methods, such as that of Gauss–Newton, converge faster for an initial estimate close to the minimum. Third, solving the problems on the coarse grid is computationally more efficient, and, in most cases, only small corrections are required at finer levels.

The pseudocode is shown in Algorithm 1.
**Algorithm 1** Multilevel registration 1:**INPUT:** Reference image (R) and test or moving image (M) 2:**for**level=minLevel⇒maxLevel**do** 3: $M_{level}\Rightarrow M$ 4: $R_{level}\Rightarrow R$ 5: $y_{level}\Rightarrow argminJ\left(y_{level}\right)$ 6: **if**
Level⊣maxLevel
**then** 7:  $y_{Level+1}⇐y_{Level}$ 8: **end if** 9:**end for**10:$y\Rightarrow y_{maxLevel}$11:**OUTPUT:** Transformation of y

## 3. Materials and Methods

The simulation studies performed in this paper were carried out using Geant4 Application for Tomographic Emission (GATE) software [73,74]. GATE is a Monte Carlo simulator focused on PET/SPECT imaging that allows the user to simulate PET scanner geometry, define different types of sources, and simulate the interaction of the particles with the detectors [73,75,76].

### 3.1. Geometry System Description

The used PET system is a ring with an 800 mm diameter, an axial and transaxial field of view (FoV) of 156 and 400 mm, respectively (see Figure 3), composed of 144 detector modules. Each module is composed of 169 pixelated LYSO scintillator crystals (7.1 g/cm^3^) with a size of 20 × 20 × 4 mm^3^. For this study, the GATE framework was used to simulate a radioactive source that emits two annihilation photons of 0.511 MeV generated at 180 °(back-to-back photons), its interaction with the crystal detector of the PET scanner, and the response of the readout electronics. A list of all coincidence pair positions, time, and energy (in list-mode format) is provided by the simulation tool.

The acquisition time was set to 60 s, and the coincidence time window was set in 4 ns.

The source activity of the phantoms varied depending on the different simulation tests, as described in following subsections, and the output was given as a list-mode dataset.

### 3.2. Phantoms

Two phantoms were designed for the experimental setup (Figure 4). The first one (cylinder phantom) consists of an external cylinder (20 mm radius and 20 mm height) filled with water. Inside, there are four cylinders (25 mm radius and 60 mm height) placed in such a way that their centers form a rectangle with a side length of 90 × 100 mm.

The second phantom (Derenzo phantom) is composed of an external cylinder filled with water (30 mm radius and 20 mm height) in which in interior, there are a total of 35 cylinders (10 cylinders with a radius of 1 mm, 10 with a radius of 1.5 mm, 6 with a radius of 1.75 mm, 6 with a radius of 2 mm and 3 with a radius of 2.65 mm).

### 3.3. Motion Tracking

As mentioned above, for frame-based motion correction algorithms, it is necessary to use an extrinsic motion tracking system (EMT). For this purpose, an optical tracking camera was simulated (provided as output positions and angles of the source at different times) following the same motion that is performed on the phantoms with GATE. In order to be more realistic, an uncertainty was introduced to the spatial axes of the camera data, which consisted of leaving around each of the positions a margin of error given by a normal distribution of mean μ = 0 mm and standard deviation σ = 0.1 mm. In the same way, the angular dimensions were assigned an uncertainty following the same distribution, but, in this case, with a deviation of 1 °.

### 3.4. Time Synchronization

Precise synchronization of the simulated motion tracking data and the simulated PET list-mode data are mandatory. In order to synchronize both files, the time marked by the camera at the beginning of the acquisition was taken as the origin indicator of times, and relative times in the list-mode files were calculated from there on.

### 3.5. Image Reconstruction

In this work, the maximum likelihood expectation maximization with ordered subsets (MLEM-OS) algorithm was used [77,78] to individually reconstruct the corresponding frames. The number of iterations for the reconstruction was five iterations with 10 subsets each. The reconstructed images have a size of 200 × 200 × 200 pixels with a 0.5 mm pixel size in all dimensions.

### 3.6. Configuration of the Simulation

For finding possible limitations in the method, a set of one (1D) and two (2D) dimensional experiments was carried out in which the intensity and size of the voxel varied.

The activity source used in the first set of simulations was the cylinder phantom defined previously in Section 3.2 (Figure 4 left).

For the 1D experiments, a sinusoidal motion was simulated given by the following Equation (Equation 2):(2)$$x\left(t\right)=100\times \sin\left(\frac{2\pi }{40}\times t\right)$$

For the 2D experiments, two motions were simulated together, a motion from top to bottom according to Bernoulli’s lemniscate (Equation (Equation 3)) and a sinusoidal motion (Equation (Equation 4)) that goes from left to right. That is, the phantom moves from left to right continuously through a sinusoid and at the same time from top to bottom, drawing the lemniscate:(3)$${\left(x^{2}+y^{2}\right)}^{2}=10000\times \left(x^{2}-y^{2}\right)$$
(4)$$x\left(t\right)=100\times \sin\left(\frac{t}{20}\right)$$

The objective of the intensity variation experiments is to observe how much the activity of the source can be decreased, in other words, to identify which is the minimum count-rate (number of coincidences) necessary to correct the motion while not compromising the quality of the final image. Several experiments in 1D and 2D were performed with different source activities of 1, 0.15, 0.10, 0.075, 0.05, 0.025, 0.015, and 0.0075 MBq of 18F-FDG (fluorodeoxyglucose). This range of activities corresponds to very low doses of radiopharmaceuticals delivered to the patient (normally, the effective dose in adults is around 400 MBq of FDG in a PET scan).

In order to be able to compare the experiments, a voxel size of half the diameter of the smallest source under study was chosen (we called this parameter *a*). In this case, a cylinder with a 50 mm diameter leads to a voxel size of a×50=0.5×50=25 mm.

The experiments related to voxel size were focused on determining the size that would maximize the quality of the image while minimizing the required reconstruction time. This was studied by varying the size of the source via the parameter *a*. For this, experiments were carried out with four values of *a*: 2, 1, 0.5, 0.25, and 0.125. This factor *a* represents the ratio between the voxel size and the smallest size of the source. As an example, a factor of a = 2 implies a voxel size two times bigger than the smallest source in the experiment. As in the previous case, in order to compare the diverse experiments, an activity of 0.15 MBq of 18F-FDG was chosen for all size experiments.

Another complementary 2D experiment was performed with the Derenzo phantom described in Figure 4 right. Here, a voxel of a = 0.5 and a source of 1 MBq of 18F-FDG activity were chosen. The objective of this experiment was to identify the spatial resolution that the algorithm can provide without losing quality in the reconstructed image.

## 4. Results and Discussion

For the validation of the EMAF motion correction algorithm, qualitatively and quantitatively analyses were performed, plotting different profiles and showing three figures of merit: peak signal-to-noise ratio (PSNR), mean square error (RMSE), and the precision of coincidence in intensities (IMP), defined, respectively, in Equations (Equation 5)–(Equation 7):(5)$${\mathrm{PSNR}}_{X,F}=10\;\log_{10}{\left({\frac{f_{\mathrm{peak}}}{\mathrm{RMSE}}}_{X,F}\right)}^{2}\;$$
(6)$${\mathrm{RMSE}}_{X,F}=\sqrt{\frac{1}{n}\sum_{i=1}^{n}{\left(F_{i}\;-\;X_{i}\right)}^{2}}$$
(7)$${\mathrm{IMP}}_{X,F}=\left(1\;-\;\frac{{\mathrm{RMSE}}_{X,F}}{\sqrt{E[F^{2}]}}\right)\times 100$$
where $E\left[F^{2}\right]=\frac{1}{n}\sum_{i=1}^{n}f_{i}^{2}$ is the second momentum of the reference image intensity distribution, and $f_{\mathrm{peak}}$ is the maximum pixel value in the reference image. $F_{i}$ and $X_{i}$ are the intensity values of the voxel *i* of reference and corrected images, respectively, and, finally, *n* is the total number of voxels.

The PSNR metric estimates the level of noise that affects the fidelity of an image representation; the IMP calculates the intensity deviations between the test image (in this case, the corrected image) and the reference image in the form of voxels. For both calculations, increasing values suggests a better concordance between the test image and the reference image. The RMSE measures the amount of error between two sets of data. In other words, it compares the corrected value of the pixel and the value of the same pixel in the reference image.

A higher IMP value implies better image matching. In the ideal case, a 100% value represents perfect motion correction, whereas a black image (or background image) yields an IMP value of (nearly) 0%.

### 4.1. Comparison with MAF

Among the frame-based motion correction methods, the multiple acquisition frame approach (MAF) method is one of the most commonly used. This method divides the PET acquisition data into multiple frames using external information provided from a tracking system and registers all the reconstructed frames to a reference image in order to obtain a motion-free reconstructed image.

In order to prove the EMAF’s capabilities, a comparison with respect to MAF was performed. This comparison was made for the 1D motion of the cylinder phantom with 0.15 MBq, and a size ratio of a = 0.5 was used in order to cut the dataset into frames. Both algorithms were compared, and the results are shown in Figure 5.

Several metrics are calculated for both algorithms in Table 3:

This result shows that EMAF is an improved version of MAF in which improvements in the frame cut method and registration lead to better results for the final reconstructed image. This is caused due to an increase in the count rate in each frame and a reduction in the intra-frame motion as a consequence of a more efficient cut method.

### 4.2. Experiments with a Variation of Source Activity

The main objective of conducting experiments with different activities is to be able to test the motion correction algorithm developed in scenarios with little statistics, which is the main limitation of frames’ motion correction approaches.

In Figure 6, the results obtained with the 1D experiments with a variation of activities are shown.

In the same figure (Figure 6), it is shown that the method is capable of correcting a distorted image without causing apparent structures caused by the motion of the source (this can be seen on the motion-affected images on the central column) while maintaining the same activity relationship of the original source.

Profiles of the previous images are depicted in Figure 7.

In Figure 7, it can be seen that the precision in intensity when comparing between the static image and the corrected image is much higher than that between the static image and the uncorrected moving image (Figure 8).

This is shown in the IMP values in both 1D and 2D. The IMP values of each of the activities are in the range of from 80 to 95%. In the cases of very low activity (0.0075 and 0.0125 MBq, respectively), these values are lower, but still higher than 70% for 1D and between 50 and 70% for 2D. All of these values are still much higher compared to those of the static and uncorrected moving image (with an IMP of around 20–30% in the case of 1D and between 10–20% in the case of 2D) for low activity.

The same trend can be observed in the RMSE. In both 1D and 2D, a higher error was obtained between the static and uncorrected images than between the static and corrected images for all cases (Figure 9). This confirms that EMAF properly corrects motion. The error increased with the source activity. This is justified by the quadratic character of the figure of merit studied.

The values of the figures of merit used in the experiments with different activities are shown for 1D and 2D in Table 4 and Table 5, respectively.

Several conclusions can be drawn from the results in Table 4 and Table 5. For most cases in the 1D and 2D experiments, the mean square error is much lower between the static image and the corrected image than between the static image and the uncorrected one. This difference increases with the activity. The RMSE in the uncorrected image remains in the range of 15–17 in both 1D and 2D motions, while, in the corrected image, this range is around 4.

In the IMP metric (Table 4 and Table 5), which controls in some way the similarity between the two images, a clear improvement between the static and corrected image versus the static and uncorrected image was found; therefore, it is quantitatively clear that the EMAF method keeps the error low and is able to reproduce the structures of the reference image.

However, for the PSNR values (Table 4 and Table 5), it can be observed that, in the low ranges of activities, the signal-to-noise ratio is higher for the static image–uncorrected image, which indicates that the corrected images present more noise. This noise is justified by a very low activity, and, as a consequence, there is a low statistic. This low statistic leads to non-optimal image reconstruction, and, therefore, the quality of the corrected image is not as good as one might expect.

However, this trend changes from 0.050 MBq for the 1D case and 0.075 MBq for the 2D case, with the signal-to-noise ratio of the static image–corrected image being greater than the uncorrected image from these two values. This result is interesting since it provides a criterion of the application range of the EMAF algorithm. In this case, a lower limit of between 0.050 and 0.075 MBq is shown, below which the algorithm presents undesirable results. Note that the lower range of limiting activities is unlikely in real scenarios (normally the effective dose in adults is 8 mSv with an activity of 400 MBq of FDG administered), so in the normal activity ranges of a PET scan, EMAF would be accurate, and its results would be acceptable.

These results are in accordance with [70], which indicates that the minimum activity for tracking point sources in each frame is 0.5 MBq. However, increasing the activity of point sources may lead to better performance, although resulting in an increase in the dose absorbed by the patient.

Ikoma et al. suggest in [79] that, to improve the precision motion detection, it is better to make the frame duration shorter. Nevertheless, the trustworthiness of the registration step depends directly on the signal–noise of the reconstructed images, especially in frames with small counts because of low activity accumulation or large radioactivity decay. However, the results presented in this paper show the overcoming of this difficulty exposed by [79] in relatively low-source activities.

### 4.3. Experiments with Different Voxel Sizes

The purpose of carrying out experiments with different voxel sizes responds to the need to know the optimal voxel size limit to gain enough statistics so that the reconstruction of the image is not affected and simultaneously does not have an excessive number of frames, which can compromise the computational effectiveness of the method.

As shown in Figure 10, the images corrected for the different sizes follow the same structure of the static image. For the voxel with a = 2 (the biggest voxel size of those that have been examined), there are halo-shaped artifacts around the cylinders that are much more evident than those of the images with a smaller voxel size.

This result is expected. For a voxel of a = 2, intra-frame motions are introduced. The same behavior is observed in one dimension, although this is not as evident as in two dimensions. This is the main reason why the method in which the voxel size can be adapted according to the size of the structures of the object of study was built.

If different profiles are generated (Figure 11), it is observed that the corrected images present the same distribution as that of the static image for all sizes, which indicates that EMAF corrects accurately. However, it can also be seen that, in all cases, intensity slightly higher than that of the static pixel is obtained. This is because the algorithm is count-dependent (it depends on the number of counts in each frame); this slight variation in intensities is caused by the re-weighting factors (weighting factors) that are introduced and that are necessary to take into account when counting the statistical differences in each. These weights take into account the time spent in a certain voxel.

At a quantitative level (Table 6), it is observed that the mean square error is always lower for the static image–corrected image pair than for the static image–uncorrected image pair. IMP values also reinforce this. Note that, even in the case where the voxel size is larger than the structure under study, the value of this metric (IMP) decreases, a direct consequence of having more intra-frame motions within the same frame. This can also be clearly seen in the signal-to-noise values. For small voxel sizes, a better signal is obtained. This trend is manifested in both 1D and 2D experiments.

### 4.4. Experiments with the Derenzo Phantom

The experiment with the Derenzo phantom aimed to quantify a possible loss of spatial resolution of the method. For this, a Derenzo phantom was created with a motion trajectory following the Bernoulli lemniscate according to Equations (Equation 3) and (Equation 4), as can be seen in Figure 12.

The frame division configuration in this case maintains 99.4% of the total events, with the residual inter-frame being around 0.6%.

At the qualitative level, it is observed that the method is able to distinguish the largest structures of the Derenzo phantom (2.65 mm in radius) from the smallest (1 mm in radius). Based on these results, it can be affirmed that EMAF accurately resolves all of the structures of the phantom. In order to show this, a profile analysis was performed in Figure 13.

At the quantitative level, the different metrics also show very satisfactory results. In this case, the PSNR figure of merit shows 45 dB for the corrected image, while the uncorrected image remains at 41 dB. The IMP also agrees with this conclusion, giving a better value for the corrected image than for the uncorrected image (57 versus 7%, respectively). The RMSE presented by the corrected image is around 3, while, for the uncorrected image, it is around 4.79. EMAF can diagnose lesions that are around 1 mm in size or larger. Taking into account that the spatial resolution of conventional PET used in the clinic is approximately a few millimeters, the uncertainty that we obtained in the motion reconstruction process (below 1 mm) can be considered more than satisfactory.

It is worth mentioning that the proposed method minimizes intra-frame motion and reduces voxel noise to levels that are considered acceptable as specified in [80,81].

In this context, Gravel et al. [82] present an inter-frame correction method with simulated data similar to that in this study; Jiao et al. [83] also develop a direct reconstruction algorithm to estimate inter-frame motion. However, none of these works addresses intra-frame motion or takes advantage of measurements from an external motion tracking device [11].

On this occasion, Jin et al. [84] suggest that, while frame-based motion correction is sufficient for small intra-frame motions (<5 mm), event-by-event motion correction has universally good performance. However, as shown in the present paper, this consideration would have no effect when applying the improvement proposed in this paper since the cases obtained through EMAF would be comparable to event-by-event motion correction approximations.

## 5. Conclusions

In this paper, a novel motion correction algorithm is proposed, which is composed of three phases (motion estimation, image reconstruction, and motion correction). The approach is based on image registration and frame acquisition of the data. Firstly, the list-mode data obtained from the scanner were sorted into quasi-static frames. Secondly, each frame was reconstructed using a built in-house MLEM-OS algorithm. Finally, all of the reconstructed frames were registered in order to obtain a final reconstructed image, taking into account that the registration had to be hyperelastic and nonlinear.

For validation of the algorithm, several experiments were designed with two phantoms (Derenzo and cylinder phantoms) in a ring with different motions (sinusoidal and Bernoulli’s lemniscate) in one and two dimensions, with a mean velocity of 9.98 mm/s and a maximum displacement of 100 mm for one dimension and a mean velocity of 15.93 mm/s with maximum displacement in an axial axis of 100 and 35.37 mm in the transaxial axis. The validation was performed qualitatively (using profiles) and quantitatively (using three figures of merit).

From the results of these experiments, several interesting conclusions can be drawn: From the experiments with different activities, it is possible to extract a lower limit of activity below which quality in the image is lost. This lower limit is much lower than the range used in this type of testing in adults, so EMAF can be a good candidate for application in clinical practice.

From the results obtained in the experiments with different voxel size ratios, it was possible to extract a range of cut sizes depending on the size of the structure under study. This fact allowed us to semi-automatically choose the optimal cut size for each of the frames, thereby enabling us to minimize the effect of inter-frame motion.

Lastly, the Derenzo experiment confirmed that EMAF can reconstruct millimeter-sized structures, meaning that the resolution achieved by the algorithm is even above the spatial resolution of most conventional scanners.

This algorithm presents some limitations. First, in order to decide a good parameter to make the division of the data in frames, an approximation of the size of the structures to study needs to be made. The better this approximation, the less noise present in the final image, as the cutting process will be possible in almost static frames. Although this approach is more flexible than other frame-based methods in which this parameter is hard-coded, it is still a limitation that needs to be taken into account. Apart from this, although the number of frames that EMAF produces is minimal, it is necessary to mention that, when large spatial motion ranges are present, it produces a high number of frames, which leads to an increase in the computational load. Despite the limitations, the EMAF algorithm is a robust enhancement of the MAF algorithm that solves the major problems and improves the performance of motion correction in a low-intensity range.

## Figures and Tables

**Figure 1 sensors-21-02608-f001:**
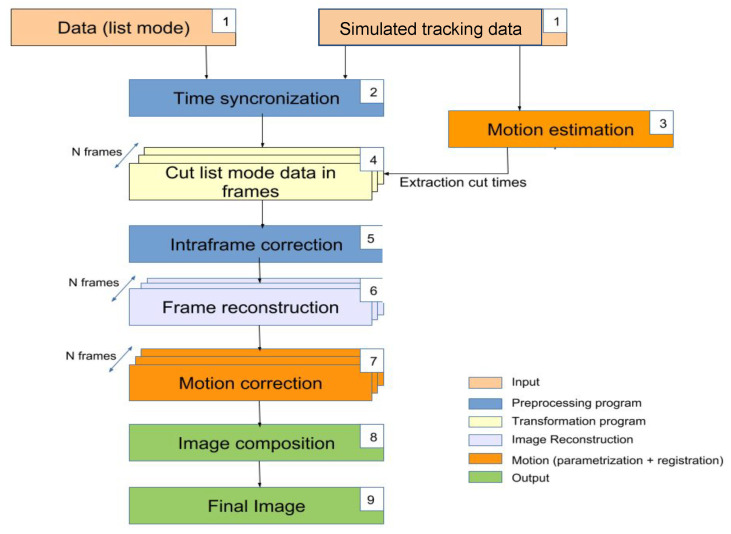
EMAF algorithm flowchart.

**Figure 2 sensors-21-02608-f002:**
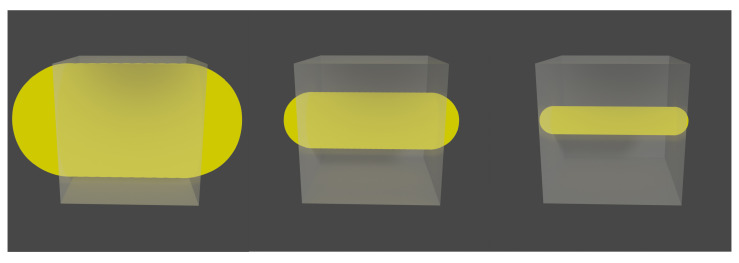
Cylindrical sources moving in a straight line inside a voxel. Sizes are 1, 2, and 4 times smaller than that of the voxel side length ((**left**) to (**right**)).

**Figure 3 sensors-21-02608-f003:**
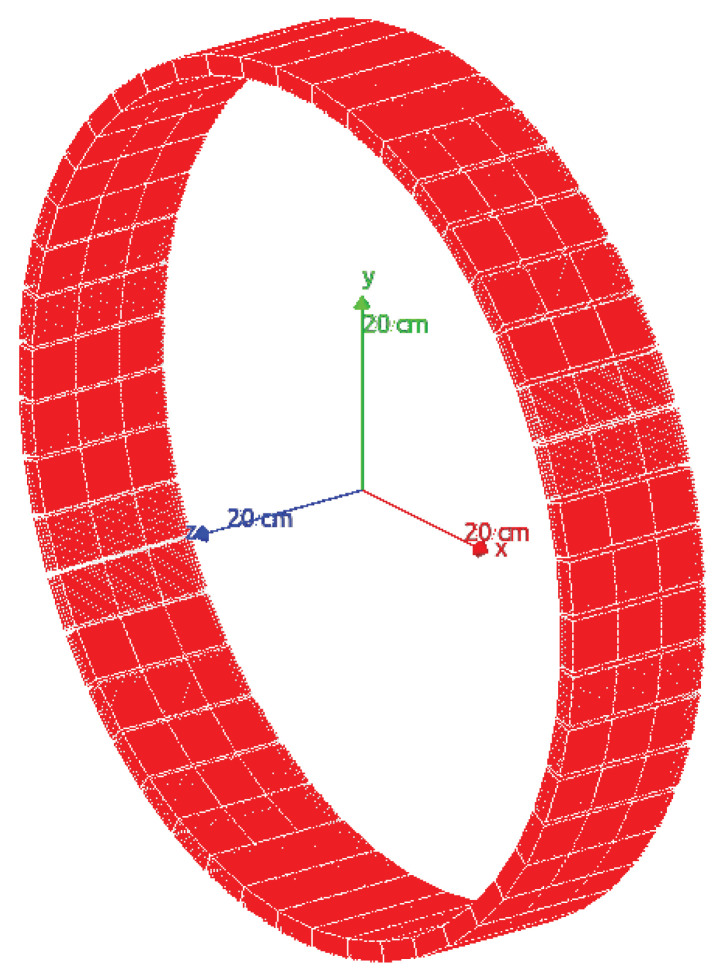
PET system object of study.

**Figure 4 sensors-21-02608-f004:**
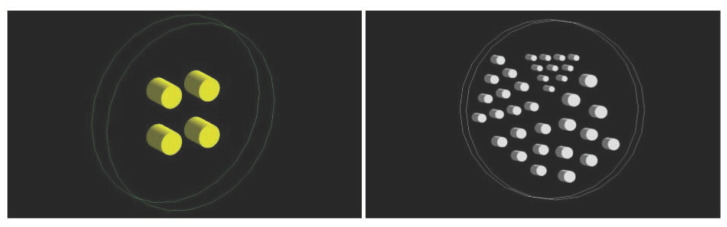
(**Left**): visualization of the cylinder phantom; (**right**): Derenzo phantom.

**Figure 5 sensors-21-02608-f005:**
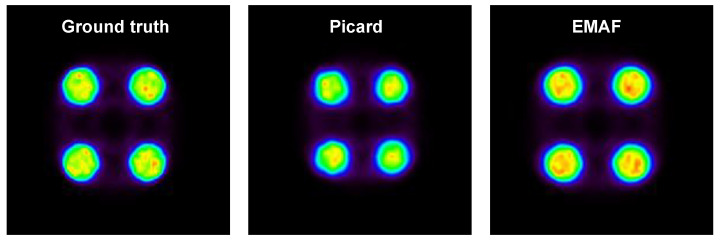
Images of the static (ground truth) image, Picard’s multiple acquisition frame approach (MAF)-corrected image, and enhanced multiple acquisition frames (EMAF)-corrected image.

**Figure 6 sensors-21-02608-f006:**
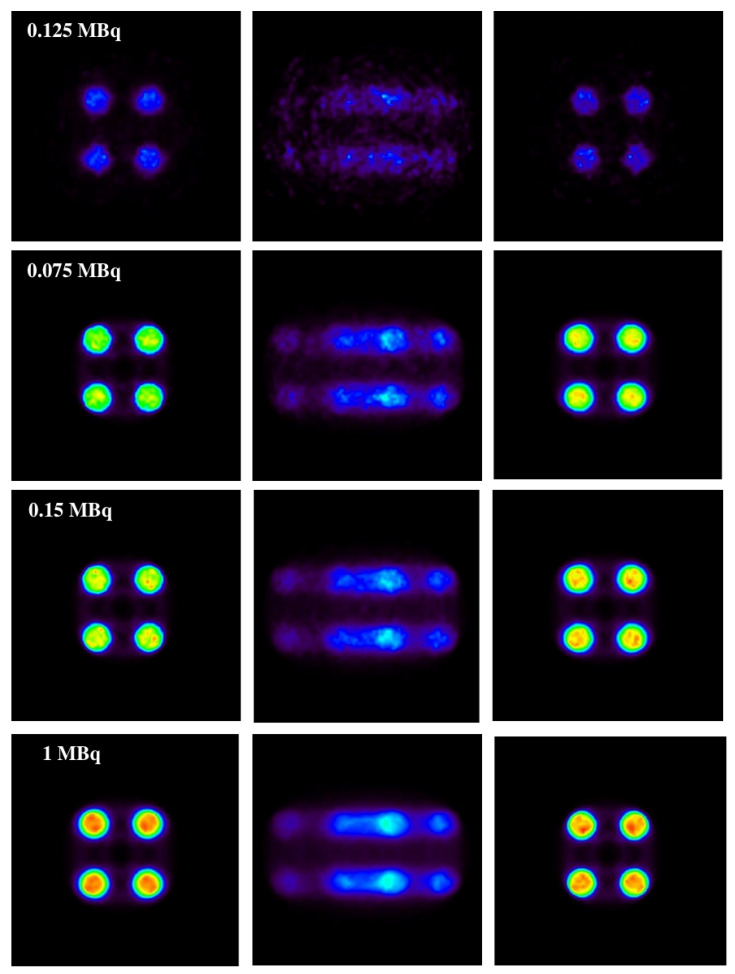
Cylindrical phantom reconstructed for different activities. From (**left**) to (**right**), static image (ground truth), uncorrected (motion-affected) image, and motion-corrected image. It is shown that the corrected image correctly preserves the activity of the original image.

**Figure 7 sensors-21-02608-f007:**
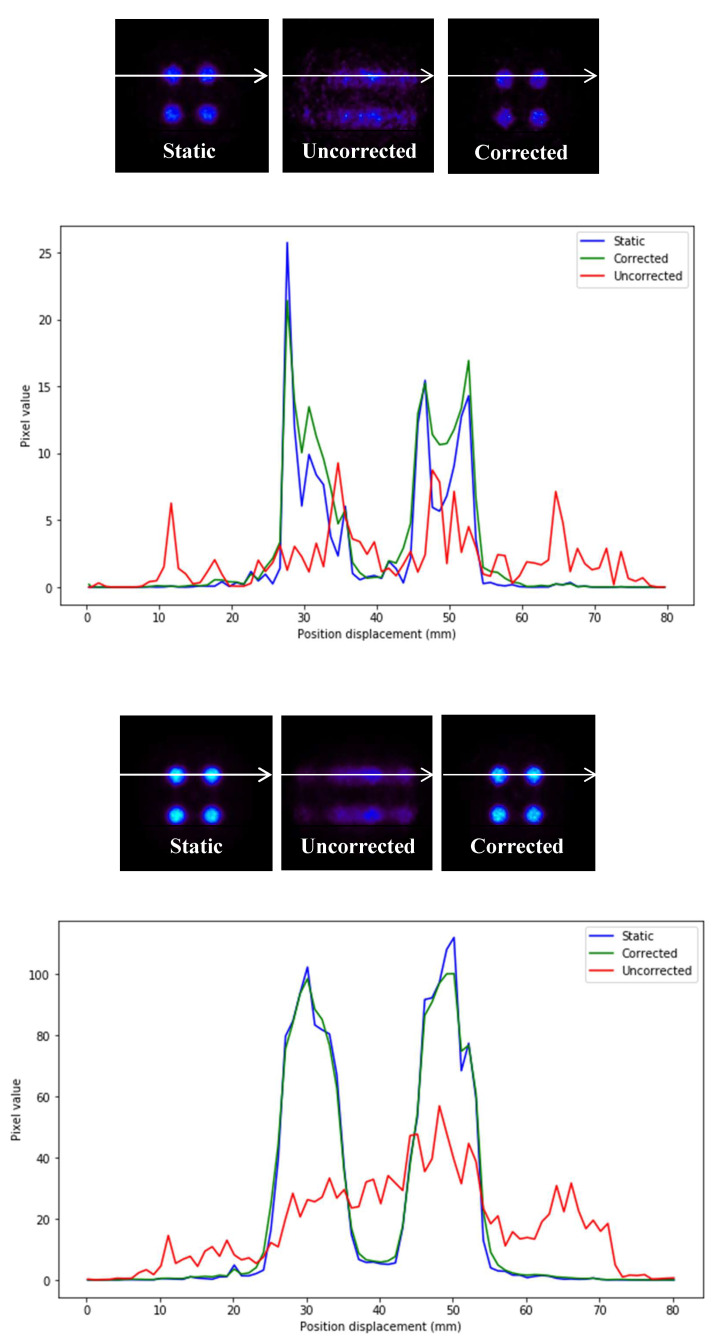
Static image, uncorrected and corrected together with their profiles (**below**) of two experiments of different activities. (**Up**): images and profiles of a low-range activity (0.0125 MBq). (**Below**): images and profiles of a high-range activity (0.1 MBq). The white line indicates the profile in the image.

**Figure 8 sensors-21-02608-f008:**
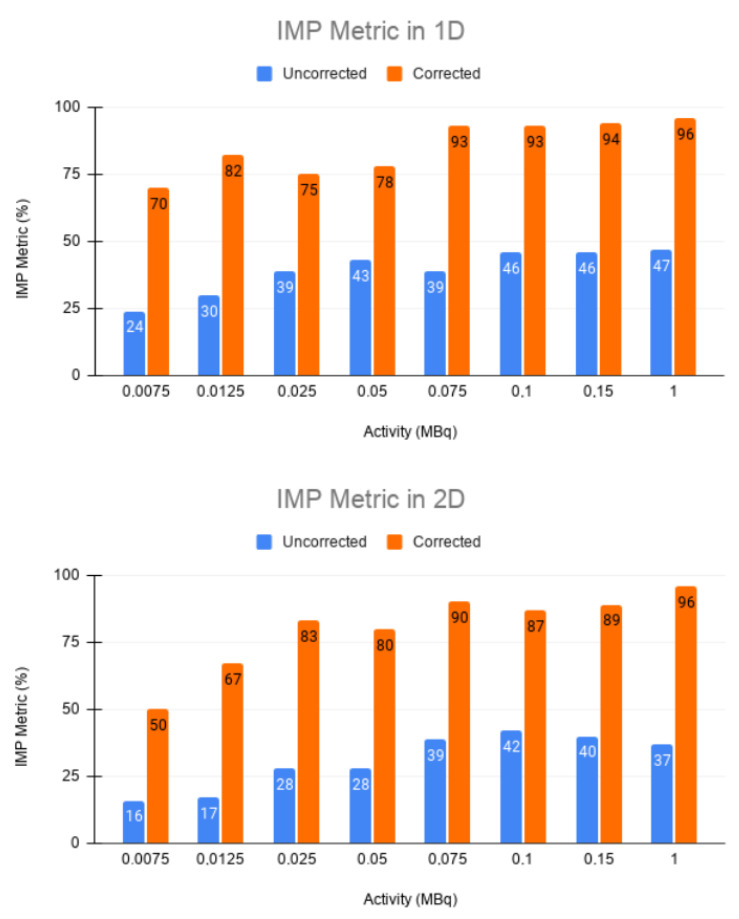
Precision of coincidence in intensities (IMP) metric values for 1D (**top**) and 2D (**bottom**) for static-uncorrected and static-corrected image pairs.

**Figure 9 sensors-21-02608-f009:**
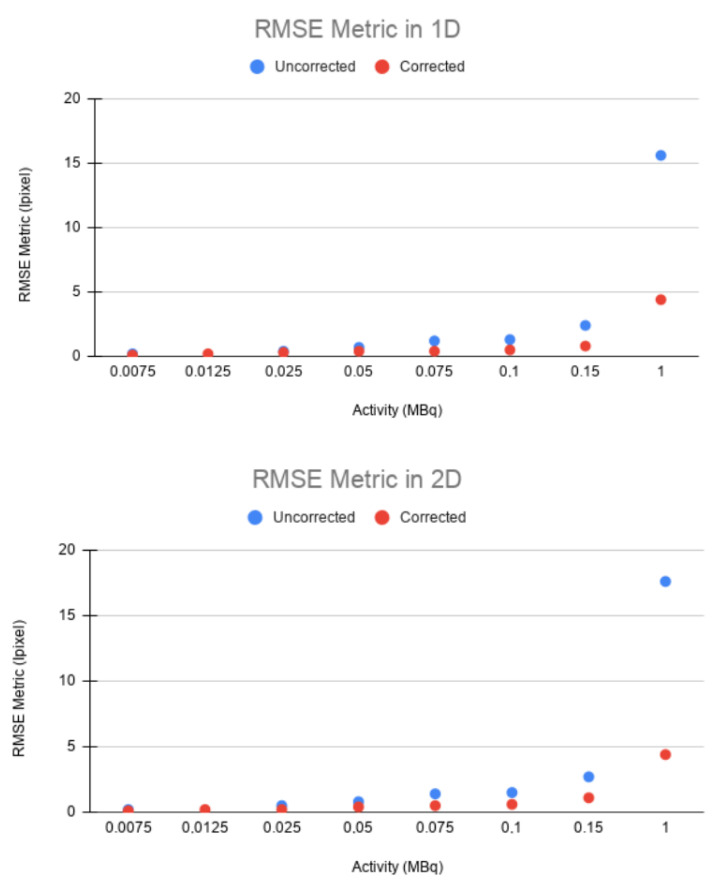
Root mean square error (RMSE) values for 1D (**top**) and 2D (**bottom**) for static-uncorrected and static-corrected image pairs.

**Figure 10 sensors-21-02608-f010:**
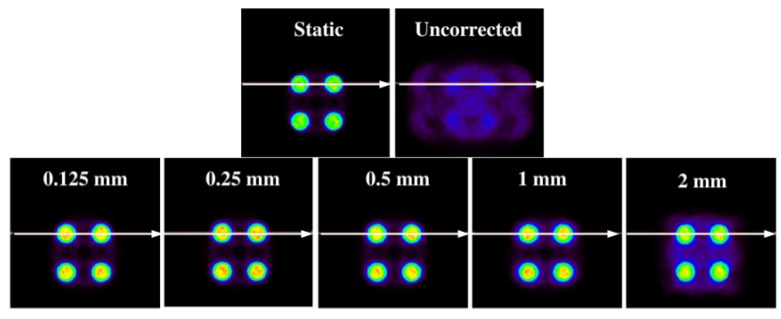
Reconstructed phantom for different voxel sizes with Bernoulli’s lemniscate motion in two dimensions. Images above: (**left**) static image; (**right**) uncorrected movie clip. Images below: images corrected for voxel size ratios of a = 0.125, a = 0.25, a = 0.5, a = 1, and a = 2, respectively. The white line indicates the line where the respective profiles were made.

**Figure 11 sensors-21-02608-f011:**
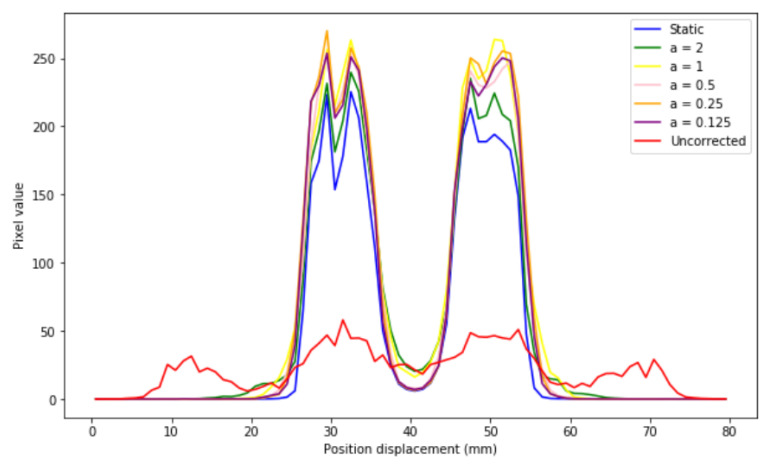
Profiles of the static, uncorrected image and the corresponding corrected images for the different voxel sizes of a = 0.125, a = 0.25, a = 0.5, a = 1, and a = 2.

**Figure 12 sensors-21-02608-f012:**
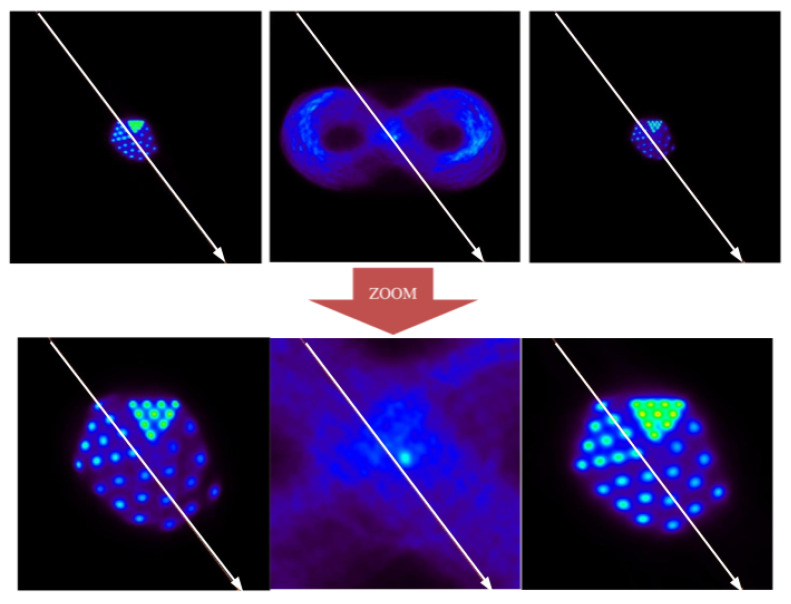
Static image, uncorrected and corrected in the 2D experiment with the Derenzo phantom. (**Top**): original size of the images. (**Bottom**): close up of the images for better visualization. The central image shows the motion followed by the phantom (Bernoulli’s lemniscate).

**Figure 13 sensors-21-02608-f013:**
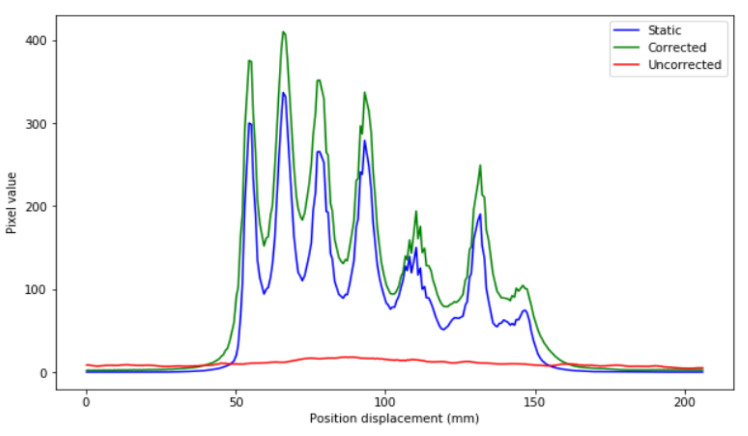
Static, uncorrected, and motion-corrected profiles of 2D motion with the Derenzo phantom.

**Table 1 sensors-21-02608-t001:** Most popular positron emission tomography (PET) image statistical reconstruction algorithms and their main features.

Software	Dimension	Features	Transformation Type
AIR	2D, 3D	Intensity	Rigid, Affine, Spline
ANTS	3D	Punctual, Intensity	Rigid, Affine, Spline, Elastic
DRAMMS	2D, 3D	Intensity	Elastic
Drop	3D	Punctual, Intensity	Rigid, Affine, Spline
ELASTIX	2D, 3D	Intensity	Rigid, Affine, Spline
FAIR	3D	Punctual, Intensity	Rigid, Affine, Spline, Elastic
Gilles	3D	Lineal, Intensity	Elastic
MIRTK	3D	Punctual, Lineal, Intensity	Rigid, Affine, Spline
NiftyReg	3D	Intensity	Spline
Plastimatch	2D, 3D	Punctual, Intensity	Rigid, Affine, Spline, Elastic
SLDIR	2D, 3D	Intensity	Elastic

**Table 2 sensors-21-02608-t002:** Similarity metrics used by different registration software

Software	Similarity Metrics
AIR	RIU, SSD, SLS
ANTS	MSD, CC, JHCT, MI, NCC, PSE
DRAMMS	CC, SSD
Drop	SAD, SADG, SSD, NCC, NMI, CR, CCGIP, HD, JRD, MI, JE, GRAD
ELASTIX	MSD, NCC, MI, NMI, DC
FAIR	SSD, NCC, MI, NGF
Gilles	NCC, SSD
MIRTK	NMI, SSD, CC, CR, JE, MI, NMI, DC
NiftyReg	NMI
Plastimatch	MSD, MI, NMI
SLDIR	MSD

**Table 3 sensors-21-02608-t003:** Comparison between Picard’s MAF and EMAF metrics.

	RMSE ± 0.1	PSNR ± 1 (dB)	IMP ± 1(%)	
MAF	2.1	42	58	
EMAF	0.6	43	94	

**Table 4 sensors-21-02608-t004:** RMSE, IMP, and peak signal-to-noise ratio (PSNR) values for a sinusoidal motion in one dimension for different source activities.

1D			
STATIC vs. MOTION			
Activity (MBq)	RMSE ± 0.1	PSNR ± 1 (dB)	IMP ± 1 (%)
0.0075	0.2	46	24
0.0125	0.2	47	30
0.025	0.4	43	39
0.050	0.7	44	43
0.075	1.2	44	39
0.100	1.3	44	46
0.150	2.4	39	46
1	15.6	38	47
STATIC vs. CORRECTED			
Activity (MBq)	RMSE ± 0.1	PSNR ± 1 (dB)	IMP ± 1 (%)
0.0075	0.1	43	70
0.0125	0.1	44	82
0.025	0.3	41	75
0.050	0.4	47	75
0.075	0.4	43	93
0.100	0.5	44	93
0.150	0.8	43	93
1	4.4	42	96

**Table 5 sensors-21-02608-t005:** RMSE, IMP, and PSNR values for “infinite”-type motion (Bernoulli’s lemniscate) in two dimensions for different source activities.

2D			
STATIC vs. MOTION			
Activity (MBq)	RMSE ± 0.1	PSNR ± 1 (dB)	IMP ± 1 (%)
0.0075	0.2	47	16
0.0125	0.2	47	17
0.025	0.5	42	28
0.050	0.8	43	28
0.075	1.4	31	39
0.100	1.5	31	42
0.150	2.7	33	40
1	17.6	34	37
STATIC vs. CORRECTED			
Activity (MBq)	RMSE ± 0.1	PSNR ± 1 (dB)	IMP ± 1 (%)
0.0075	0.1	42	50
0.0125	0.2	40	67
0.025	0.2	40	83
0.050	0.4	39	80
0.075	0.5	41	90
0.100	0.6	39	87
0.150	1.1	37	89
1	4.1	40	96

**Table 6 sensors-21-02608-t006:** RMSE, IMP, and PSNR values for a sinusoidal motion in one and two dimensions for different voxel size ratios.

**1D**			
Voxel Size Ratio (a)	RMSE ± 0.1	PSNR ± 1 (dB)	IMP ± 1 (%)
0.125	0.9	40	93
0.25	0.9	40	93
0.5	0.8	43	93
1	0.8	39	93
2	1.1	37	90
Static-Uncorrected	2.4	39	46
**2D**			
Voxel Size Ratio (a)	RMSE ± 0.1	PSNR ± 0.1 (dB)	IMP ± 1 (%)
0.125	1.1	39	89
0.25	1.2	39	87
0.5	1.1	37	89
1	1.2	38	87
2	1.1	34	89
Static-Uncorrected	2.7	33	40

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
