# Peer review of "Simulation Study of a Frame-Based Motion Correction Algorithm for Positron Emission Imaging"

_sensors, 2021, doi:10.3390/s21082608_

Round 1

Reviewer 1 Report

The paper promises to provide a novel technique for image correction for PET systems. I believe this study could be interesting to experts working with such systems. However, the study needs to be motivated better (one idea to realize this is to point out the advantages for clinical experts as indicated in the conclusions). Moreover, the paper is currently very hard to read and follow as it is full of grammatical mistakes. One example is the continuous misuse of singular and plural throughout the whole document. I recommend a language check by a professional before a real peer-review is possible.   In addition to the language, the following issues should be addressed:     title - avoid acronyms (PET) in title, also there seems to be an article missing   abstract - some acronyms are not defined       line 98: give examples that support this statement. line 105: Please state the advantages of your estimation and correction algorithm. Moreover, explicity say why the above listed algorithms are disadvantageous for your case.   Algorithm 1: please align correctly   Section 3.6, 4.1 and 4.3: When you speak about experiences (appears several times in named sections but also other parts of the paper) do you mean experiments? The logic of the paper is very hard to follow because of the many typos and grammar mistakes.   Figure 7: This is a nice result. Please add also a figure with other algorithms, so that the reader can see that your algorithm is superior.   line 339: adding the cylinders up leads to 35 not 25   line 387: It is kind of expected that the corrected image would have less error than the uncorrected one. It would be more interesting if you compare an image corrected with your algorithm with an image corrected by another algorithm.   Figure 11: the orange line is hardly visible.   Table 5: the caption talks about one dimension but there is also 2D in the table.   line 483: Please state explicitly the levels that considered acceptable.   line 515-516: This is a nice conclusion. Please elaborate. You might add this also to the abstract and introduction to motivate this paper better.   Limitations are missing from the conclusions.  

Reviewer 2 Report

The first comment I'd like to make is that the manuscript needs a thorough review of the English language by a speaker with a very good level of English. The way it is written in the present form is very poor, and it makes difficult to understand the information it is trying to convey.

The second general comment is the following. In the introduction you state that your method is an improvement of the Picard's method. I miss a clear comparison of the performance of this method with the previous Picard's method. The discussion of the results shows that the method works, but doesn't show that improves on the state of the art. Please, discuss it more quantitatively.

Lines 169-170: I don't understand why the example of how to assign an index to each voxel is relevant in the discussion of the method. Also, why in Figure 3 - right aren't the voxels contiguous, while in the left image they are? In the text the grid is said to be "equidistributed and equispaced".

Section 3.1: I miss a description of what is being actually simulated: are the optical photons tracked and propagated through the detector until reaching a photosensor and the positions/times of the coincidences reconstructed? If not, how do you extract the list-mode dataset? Please explain it.

Section 3.6: why do you call the simulations experimental setups? I think it's misleading, you could simply call them different configurations of the simulation.

Lines 302-305: here you are repeating what has been already defined in section 3.2. Why don't you just use the term "cylinder phantom" defined there?

Lines 318 and following. The meaning of the "a" parameter is confuse. In lines 329-330 it seems that a is the ratio between the voxel size and the smallest size of the source in the simulation. However, in line 318 it seems that a is 50 mm. If the definition of "a" changes in the two configurations, consider using a different letter for clarification. Please, correct the notation and explain it better.

Lines 337-341: same as above: don't repeat the description of the Derenzo phantom.

Equations 5, 6 and 7 and lines 357-360 should appear immediately after line 349.

line 380: looking at Fig. 9, I'd say that IMP using corrected images for the 2D case is between 50 and 70% for low activity, not 40-50% as you quote in the test. Am I interpreting the plot correctly?

Figure 10, caption: the caption says IMP metric value, but the plot seems to show RMSE metric.

All tables: what is the unit dB in the PSNR figure of merit? Isn't it a ratio (therefore a  number)?

For low activities the difference between corrected and uncorrected cannot be appreciated for some metrics, being within error (I understand that the error is what appears in the titles of the columns of Table 4 and 5). Instead, in the text (lines 392-395) you state that for any activity the difference is huge. It seems contradictory to me, please, rephrase consistently.

Section 4.2, experiment with different voxel sizes.
In Table 5, the PSNR figure of merit seems to be better for uncorrected images in the 2D case and of the same error (or only slightly worse) in the 1D case. Can you comment on it? How it is related with the better performance of IMP or RMSE? Here the activity of the source is 0.15 MBq, well above the threshold you found in the previous case, starting from which the PSNR was better for corrected than uncorrected image. It seems contradictory to me. Please, comment.
The first voxel size listed in Table 5 is 125, I understand it's a typo: please, correct it.

Typos and misspellings (just a few that caught my attention, however, the whole document should be reviewed carefully by an English native level speaker)
line 14 and line 122: misspelling of Picard.
Table between line 247 and line 248: there's a mix of nomenclature between Spanish (nivel) and English (level), please, fix it.
line 260: adquisition -> acquisition
line 446: "is correcting accurately and correctly": to correct correctly is a pleonasm, drop it.
line 453: Table 6: it refers to Table 5, actually.

Round 2

Reviewer 1 Report

I believe that this manuscript has potential and the authors made some efforts to improve the earlier version. However, my main concern with the paper (the language and grammar) was not fixed. It is very hard to review this paper in its current form as several sentences do not make any sense. Moreover, as already pointed in my previous review, singular and plural are continuously mixed throughout the whole document. There are numerous professional services offering language checks for academic papers. I suggest that the authors use one of those.   Here I am giving some examples of incomprehensible sentences. Note that this is by no means a complete list. The manuscript is full of those.   Abstract: "The results obtained shows that the method minimise the intra-frame motion, improve the signal intensity over the background in comparison with other literature methods, produce excellent values of similarity with the ground-truth (static) image and is able to find a limit in the patient injected dose the when some prior knowledge of the lesion is present."   line 66: "System matrix modelling [38] this technique uses the data in sinogram space and impose a fixed transformation field derived from the information derived from the PET / CT, instead of independently reconstructing the individual frames and the sum all of them up in image space, resulting in an improvement of the noise characteristics in the final images. This approach have some disadvantages:"   line 81: "Used conservatively, however, it has its interesting results as [43–45]."   line 83: "Event rebinning [46], this method estimate the motion and transfer that information to the spatial coordinates of each line-of-response (LOR) rebinning them to a new position accordingly to the patient motion and then reconstructing the image stat- ically."     line 187: 1) it allows to group in the same frame motions occurring in the same spatial region, which increases the statistical information, being this one of main drawback of the original MAF version [51]; and 2) it does not need to prefix a priori the number of possible frames, which allows to obtain a greater or lesser accuracy depending on other voxel size of implementation being used."
